# Product lambda-doublet ratios as an imprint of chemical reaction mechanism

P.G. Jambrina[1], A. Zanchet[1,2], J. Aldegunde[3], M. Brouard[4] & F.J. Aoiz[1]

In the last decade, the development of theoretical methods has allowed chemists to reproduce and explain almost all of the experimental data associated with elementary atom plus diatom collisions. However, there are still a few examples where theory cannot account yet for experimental results. This is the case for the preferential population of one of the $\Lambda$-doublet states produced by chemical reactions. In particular, recent measurements of the OD($^2\Pi$) product of the O($^3$P) + D$_2$ reaction have shown a clear preference for the $\Pi(A')$ $\Lambda$-doublet states, in apparent contradiction with *ab initio* calculations, which predict a larger reactivity on the $A''$ potential energy surface. Here we present a method to calculate the $\Lambda$-doublet ratio when concurrent potential energy surfaces participate in the reaction. It accounts for the experimental $\Lambda$-doublet populations via explicit consideration of the stereodynamics of the process. Furthermore, our results demonstrate that the propensity of the $\Pi(A')$ state is a consequence of the different mechanisms of the reaction on the two concurrent potential energy surfaces

[1] Departamento de Química Física I (Unidad Asociada CSIC), Facultad de Ciencias Químicas, Universidad Complutense de Madrid, 28040 Madrid, Spain. [2] Instituto de Fisica Fundamental (CSIC), Serrano 123, 28006 Madrid, Spain. [3] Departamento de Química Física, Facultad de Ciencias Químicas, Universidad de Salamanca, 37008 Salamanca, Spain. [4] Chemistry Research Laboratory, The Department of Chemistry, University of Oxford, 12, Mansfield Road, Oxford OX1 3TA, UK. Correspondence and requests for materials should be addressed to F.J.A. (email: aoiz@quim.ucm.es).

Chemists are keen to describe chemical reactions in terms of the motion of billiard balls on a more or less complex quantum electronic landscape, the potential energy surface (PES). However, this picture is not always valid and quite often several PESs have to be considered, potentially giving rise to non-adiabatic effects that may have a decisive influence on the dynamics. When multiple PESs participate in a reaction, it is not possible to disentangle experimentally the contribution of each of the competing surfaces, answering the question of which of them is more/less reactive and why. The presence of multiple PESs correlating reactants and products leads to open shell molecules in which the rotational levels may be split into spin–orbit states and, in turn, each of them into two nearly degenerate $\Lambda$-doublet levels that can be spectroscopically resolved due to different selection rules. In spite of the tiny energy difference between the $\Lambda$-doublet pair of states, a clear preference towards one of them is observed in many chemical processes[1–6], including inelastic and reactive collisions, and molecular photodissociation, and has even been postulated as the origin of OH astronomical masers[7,8]. As pointed out by several authors[8–17], the $\Lambda$-doublet population acts as a fingerprint to unravel the symmetries of the surfaces involved in the process, such that the propensity for one of the manifolds reflects the competing reactivity on concurrent PESs and addresses the question of where the electrons go when the reaction takes place[15,17]. However, a general, clear-cut relationship between them has not yet been demonstrated.

Collisions leading to NO($^2\Pi$) and OH($^2\Pi$) are prototypical for the study of $\Lambda$-doublet propensities. Recent experiments by Minton, McKendrick and coworkers[5,6] have determined the OD($X^2\Pi$) state-to-state $\Lambda$-doublet population ratios for O($^3$P) + D$_2$ collisions. Regardless of the collision energy and final vibrational state, they consistently found a significantly larger population of the $\Pi(A')$ $\Lambda$-doublet state compared with the $\Pi(A'')$ one, where the labelling of the states refers to the location of the singly occupied orbital in the rotation plane of the diatom, $\Pi(A')$, or perpendicular to it, $\Pi(A'')$, in the limit of high products rotational states $j'$ (refs 14,16,18). This result seems to contradict the theoretical results, which would predict a preference for $\Pi(A'')$ under the assumption that collision on two concurrent PESs of symmetry $^3A'$ and $^3A''$ would only form $\Pi(A')$ and $\Pi(A'')$ $\Lambda$-doublet states, respectively. This simple assignment is supported by the rationale that for direct, sudden collisions, the products 'remember' the collision conditions and hence there should be a close relationship between both symmetries. However, a general procedure for connecting the reactivity on concurrent PESs with the $\Lambda$-doublet population has yet to be achieved.

In what follows, we present a method that connects the reactivity on the $A'$ and $A''$ PESs with the populations of the respective $\Lambda$-doublet states, through the explicit consideration of the reaction stereodynamics. This method is capable of explaining and reproducing the experimental OD($X^2\Pi$) $\Lambda$-doublet population ratios measured for the O($^3$P) + D$_2$ reaction. The present theory also allows us to connect the predicted $\Lambda$-doublet propensities with the reaction mechanisms on each of the concurrent PESs.

## Results

### $\Lambda$-doublet populations and reactivity on $A'$ and $A''$ PESs.
We will start by invoking conservation of the reactive flux, which implies that the population of the two $\Lambda$-doublet states and the cross-sections on the $A'$ and $A''$ PESs are related by

$$\sigma_{v'j'}(\Pi(A')) = W_{A'}\sigma_{v'j'}(A') + (1 - W_{A''})\sigma_{v'j'}(A'') \quad (1)$$

$$\sigma_{v'j'}(\Pi(A'')) = (1 - W_{A'})\sigma_{v'j'}(A') + W_{A''}\sigma_{v'j'}(A''), \quad (2)$$

where $\sigma_{v'j'}(A')$ and $\sigma_{v'j'}(A'')$ are the rovibrational-state resolved cross-sections on the two respective PESs, and $W_{A'}$ and $W_{A''}$ represent the 'correction factors' to obtain the $\Lambda$-doublet cross-sections for a given $v',j'$ rovibrational state. As commented above, in the sudden limit, the flux ending on $A'$ and $A''$ PESs are assigned to $\Pi(A')$ and $\Pi(A'')$ states, respectively, which is equivalent to setting $W_{A'} = 1$ and $W_{A''} = 1$.

For a given nuclear geometry, the weights $W_{A'}$ and $W_{A''}$ are the square of the coefficients that define the expansion of the D–OD asymptotic electronic wavefunctions in terms of the $\Lambda$-doublet molecular wavefunctions $\varphi[\Pi(A')]$ and $\varphi[\Pi(A'')]$[15],

$$\psi_{A'} = a_1^{A'}\varphi[\Pi(A')] + a_2^{A'}\varphi[\Pi(A'')] \quad (3)$$

$$\psi_{A''} = a_1^{A''}\varphi[\Pi(A')] + a_2^{A''}\varphi[\Pi(A'')] \quad (4)$$

These coefficients are related to the dihedral angle between the three-atom plane and the OD molecular plane[15]. This angle connects the symmetry of the PES to that of the $\Lambda$-doublet state and, in the high $j'$ limit, can be identified with $\theta_{j'u}$, the angle between the rotational angular momentum, $\mathbf{j'}$, perpendicular to the OD rotation plane and the vector $\mathbf{u}$ perpendicular to the three-atom plane[19] (see Supplementary Fig. 1). For the $A'$ PES, the singly occupied orbital lies in the triatomic plane and, hence, $a_1^{A'} = \cos\theta_{j'u}$, and $a_2^{A'} = \sin\theta_{j'u}$. Conversely, for the $A''$ PES, the orbital lies perpendicular to the triatomic plane, leading to $a_1^{A''} = -\sin\theta_{j'u}$, and $a_2^{A''} = \cos\theta_{j'u}$.

To obtain the weights $W_{A'}$ and $W_{A''}$, one just needs to average $\cos^2\theta_{j'u}$ over one rotational period for calculations on the $A'$ and $A''$ PES, respectively, as indicated in the Methods section. This is straightforward in the quasiclassical trajectories (QCT) framework[19], where $\theta_{j'u}$ can be computed at every step of the trajectory. In a pure quantum mechanical (QM) context, the equivalent magnitude would be $\langle j_u'^2\rangle/(j(j+1))$, where $j_u'^2$ is the projection of the rotational angular momentum along the $\mathbf{u}$ vector.

A crucial finding, which can be demonstrated using either QCT or QM arguments (see Methods section for a detailed derivation), is that $\langle\cos^2\theta_{j'u}\rangle_{\text{rot}}$, the average value of the square angle cosine is related to the helicity, $\Omega'$, the projection of $\mathbf{j'}$ on the products recoil direction ($\mathbf{k'}$), through the expression:

$$\langle\cos^2\theta_{j'u}\rangle_{\text{rot}} = 1 - \left|\frac{\Omega'^2}{j'(j'+1)}\right|^{1/2} \quad (5)$$

### $\Lambda$-doublet populations and the reaction stereodynamics.
Equation (5) has very important implications: $W_{A'}$ and $W_{A''}$ for a given rovibrational state depend only on the distribution of the helicities and, in general, will differ because such distributions reflect the mechanisms on the concurrent PESs, which can be different. This means that equations (1) to (5) can be used to: (i) determine $\Lambda$-doublet populations also in a purely QM context, for which $\Omega'$ is well defined, and (ii) relate the $\Lambda$-doublet populations to the reaction mechanism (see below).

The average value of $|\Omega'|^2$ can be determined from the product rotational alignment moment, $a_0^{(2)}(j')$, which contains the essential information about the alignment of $\mathbf{j'}$ with respect to the product recoil velocity, and is given by[20] (see Supplementary Note 1 for further details)

$$a_0^{(2)}(j') = \frac{\sum_{\Omega'}\sigma_{v'j'}(\Omega')\langle j'\Omega', 20 \,|\, j'\Omega'\rangle}{\sigma_{v'j'}}$$

$$= \frac{\sum_{\Omega'}\sigma_{v'j'}(\Omega')[3\Omega'^2 - j'(j'+1)]}{2C\sigma_{v'j'}}, \quad (6)$$

where $\sigma_{v'j'}(\Omega')$ is the cross-section resolved in $(v',j',\Omega')$, $\langle ::,20|::\rangle$ is the Clebsch–Gordan coefficient. $C = [j'(j'+1)(j'-1/2)(j'+3/2)]^{1/2}$, which for high-enough $j'$ is $\approx j'(j'+1)$. The average value of $\Omega'^2$ for a given $j'$ is

$$\langle \Omega'^2 \rangle = \frac{\sum_{\Omega'} \sigma_{v'j'}(\Omega')\Omega'^2}{\sigma_{v'j'}} = \frac{2C\,a_0^{(2)}(j')}{3} + \frac{j'(j'+1)}{3}, \quad (7)$$

leading to the following expression for $W_{A'}$,

$$W_{A'} = 1 - \left[\frac{\langle \Omega'^2 \rangle}{j'(j'+1)}\right]^{1/2} \approx 1 - \left[\frac{2}{3}a_0^{(2)}(j') + \frac{1}{3}\right]^{1/2}, \quad (8)$$

in which the polarization moments, $a_0^{(2)}(j')$, have been calculated on the $A'$ PES. Identical expressions hold for $W_{A''}$ when the $a_0^{(2)}$ alignment moment is calculated on the $A''$ PES are used. Equation (8) has one important consequence: the stereodynamics of the products—specifically, the $\mathbf{k}'$–$\mathbf{j}'$ correlation—relates the $\Lambda$-doublet populations to the reactivity on the $A'$ and $A''$ PESs. Classically, $a_0^{(2)}$ lies in the range $[-1/2, 1]$, whereas its QM limiting values depends on $j'$. Negative values of $a_0^{(2)}$, close to its lower limit, correspond to $\mathbf{j}' \perp \mathbf{k}'$ and $|\Omega'| \approx 0$, whereas positive values, close to 1, imply that $\mathbf{j}' \| \mathbf{k}'$ and $|\Omega'| \approx j'$. According to equation (8), weight factors close to 0 are associated with $a_0^{(2)} \approx 1$; that is, products on the $A'$ PES would appear as the $\Pi(A'')$ $\Lambda$-doublet state and *vice versa*. When $a_0^{(2)} \approx -1/2$, the weight factor tends to 1 and products on the $A'$ PES would correspond to the $\Pi(A')$ $\Lambda$-doublet state.

**$\Lambda$-doublet ratio can be predicted for the $O(^3P) + D_2$ reaction.** Aiming to test the method and to try to reproduce the experimental results of Minton and colleagues[5,6], we carried out adiabatic time-independent QM and QCT calculations following the procedures described in refs 21–23 using a new set of $^3A'$ and $^3A''$ PESs.

The adiabatic QM state-to-state reactive cross-sections for the $O(^3P) + D_2$ reaction at $E_{coll} = 25\,kcal\,mol^{-1}$, one of the energies of the experiments carried out by Minton and colleagues[5], are

represented on the top left panel of Fig. 1. These results show that, although for low $OD(v'=0, j')$ rovibrational states the $A'$ PES is as reactive as the $A''$ one, for $j' > 12$ the integral cross-sections (ICS) on the $A''$ PES are considerably larger than those on the $A'$ PES. This is not surprising because, although both PESs have the same barrier height, the potential energy rises faster with the bending angle for the $A'$ electronic state than for the $A''$ state[24], that is, the 'cone of acceptance' is broader on the $A''$ PES. The inclusion of non-adiabatic couplings in the dynamics does not change this picture, as both trajectory surface hopping[25] and non-adiabatic QM calculations[26,27] also indicate larger reactivities on the $A''$ PES.

Before discussing the other panels of Fig. 1, it is pertinent to inspect the integral alignment moments, $a_0^{(2)}$, which are shown in Fig. 2 as a function of the rotational state for $v'=0$. The differences between the values and the trends in $a_0^{(2)}(j')$ on the two PESs are clear to see and indicate that very different stereodynamics are at play on the two surfaces. For reaction on the $A'$ PES, $\mathbf{j}'$ is strongly polarized perpendicular to the recoil direction, $\mathbf{k}'$, for essentially all $j'$ states and in some instances ($j' = 15-17$) the $a_0^{(2)}$ values are very close to their limiting negative value. In stark contrast, on the $A''$ PES, $\mathbf{j}'$ is almost unpolarized for $j' \leq 15$, with small $a_0^{(2)}$ values close to the isotropic limit, $a_0^{(2)} = 0$. With increasing $j'$ above 15, $a_0^{(2)}$ becomes gradually more negative approaching the values found on the $A'$ PES. The values of $a_0^{(2)}(j')$ on the two PESs for $v'=1$ and $25\,kcal\,mol^{-1}$ and $v'=0$ at $20\,kcal\,mol^{-1}$ are shown in Supplementary Figs 2 and 3.

Inserting the values of the alignment moments calculated on both PESs into equation (8) yields the weight factors, $W_{A'}$ and $W_{A''}$. They are shown in the bottom panel Fig. 1 for $v'=0$ at $E_{coll} = 25\,kcal\,mol^{-1}$. As can be seen, for $j' < 15$, $W_{A''} < 0.5$, which, in effect, means that $>50\%$ of the reactivity on the $A''$ PES is 'transferred' to the $\Pi(A')$ $\Lambda$-doublet state. In contrast, as a result of the consistently fairly negative values of the alignment parameters, $W_{A'}$ is always $>0.6$ and in some cases is a large as 0.80. In consequence, the relative 'transfer' of reactivity from $A'$ to $\Pi(A'')$ is much less significant than that found from $A''$ to $\Pi(A')$.

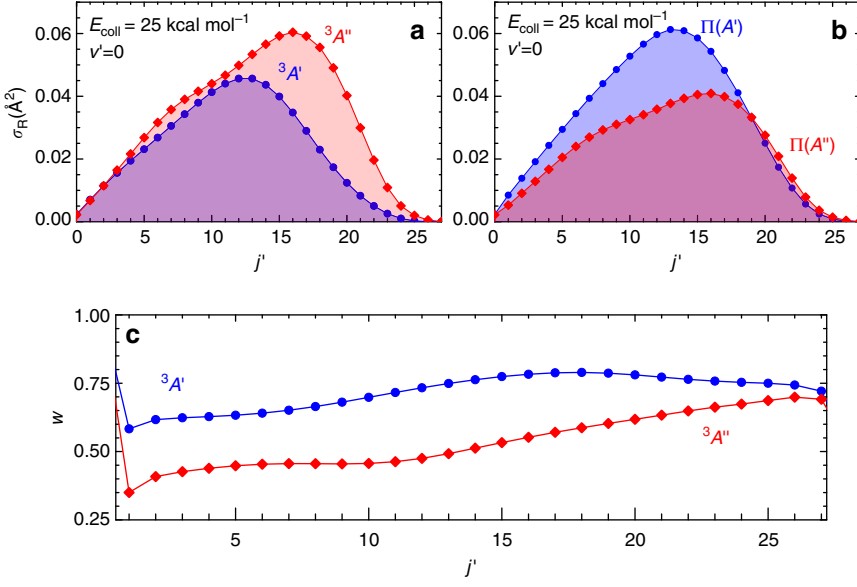

**Figure 1 | QM weighting factors to determine the $\Pi(A'')$ and $\Pi(A')$ populations.** (**a**) Reactive cross-sections calculated on the $A'$ and $A''$ PES, which, in the sudden limit, would represent the $\Pi(A')$ and $\Pi(A'')$-state resolved cross-sections. (**b**) The reactive cross-sections calculated for the two $\Lambda$-doublet levels once the respective weights have been incorporated. (**c**) Evolution of the weights for the conversion from the reactivity on the $A'$ and $A''$ PESs to the $\Lambda$-doublet populations as a function of the final rotational states. The data were obtained from the QM reaction cross-sections for the $O(^3P) + D_2$ reaction at $E_{coll} = 25\,kcal\,mol^{-1}$.

Therefore, after correction, the relative population on the $\Pi(A')$ state is significantly enhanced. The resulting $\Lambda$-doublet populations are depicted in the right panel of Fig. 1. Quite remarkably, the situation is the reverse of that found for the reactivity on the respective PESs: the $\Pi(A')$, $\Lambda$-doublet state is considerably more populated than the $\Pi(A'')$ state for low $j'$. In particular, for $j' = 12$, $\sigma(\Pi_{A'}) = (3/2) \times \sigma(\Pi_{A''})$. At higher $j'$

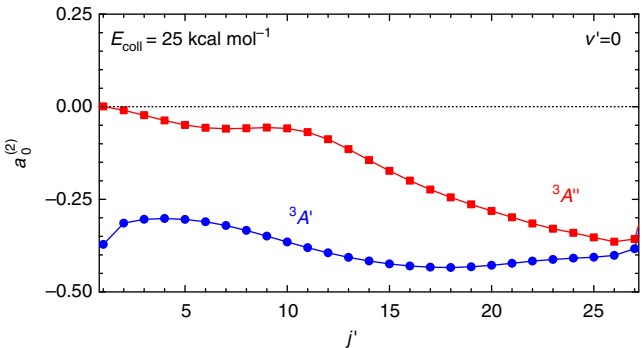

**Figure 2 | Product QM alignment parameter $a_0^{(2)}$ referenced to k′ which defines the z axis.** The alignment moment $a_0^{(2)}$ is given by the average value $C^{-1}\langle 3\hat{j}_z'^2 - \hat{j}'^2\rangle/2$, where $\hat{j}'^2$ and $\hat{j}_z'$ are the rotational angular momentum operators and $C$ is the constant that appears in equation (6). Calculations are presented for $O(^3P) + D_2(v = 0, j = 0) \rightarrow OD(v' = 0, j') + D$ at $E_{coll} = 25\,kcal\,mol^{-1}$ on the $A'$ and $A''$ PESs. Although the product rotational angular momentum, **j′**, on the $A'$ PES is strongly polarized perpendicular to **k′** (negative $a_0^{(2)}$ values), for the $A''$ PES the distribution of **j′** is largely isotropic for low $j'$.

values ($j' > 18$), the populations of the two $\Lambda$-doublets are very similar.

In Fig. 3, the experimental $\Lambda$-doublet population ratios measured[5] at 25 kcal mol$^{-1}$ are compared with the present QM (left panels) and QCT (right panels) calculations for the $v' = 0, 1$ manifolds. All the results are plotted against $N' = j' + 1$, where $j'$ and $N'$ stand for the nuclear (closed shell) and total (apart from spin) rotational angular momentum, respectively. For each case, two series of results are shown: (i) the ratio of the ICSs on the $A'$ and $A''$ (labelled as 'QCT' and 'QM') where $W_{A'}$ and $W_{A''}$ are implicitly set to 1, and (ii) the ratio of the populations on the $\Pi(A')/\Pi(A'')$ using equations (1) and (2) with the correction factors included. For the latter results (labelled as 'corr-QM' and 'corr-QCT'), the $W_{A'}$ and $W_{A''}$ factors are calculated according to equation (8). It is evident that the uncorrected QCT and QM results cannot account for the experimental $\Lambda$-doublet ratios and, regardless of $j'$, predict larger populations on the $\Pi(A'')$ state, in striking disagreement with the experimental results. In contrast, the corrected results reproduce fairly well the experimental values. In particular, the 'corr-QM' results are within the experimental error bars for most of the final states shown, particularly for OD($v' = 1$).

As shown in Fig. 4, similar agreement between experimental[6] and theoretical results is obtained at $E_{coll} = 20\,kcal\,mol^{-1}$. At even lower collision energies, $E_{coll} = 15\,kcal\,mol^{-1}$, the agreement between the corrected QCT and experimental results is not as good, probably because the collision energy is just above the barrier. In fact, no trajectories were found for $N' > 13$, whereas the QM and experimental data populate up to $N' = 21$. The corrected QM results remain in good agreement with the experiments at this low collision energy. It is worth noticing that although our corrected results predict quantitatively the

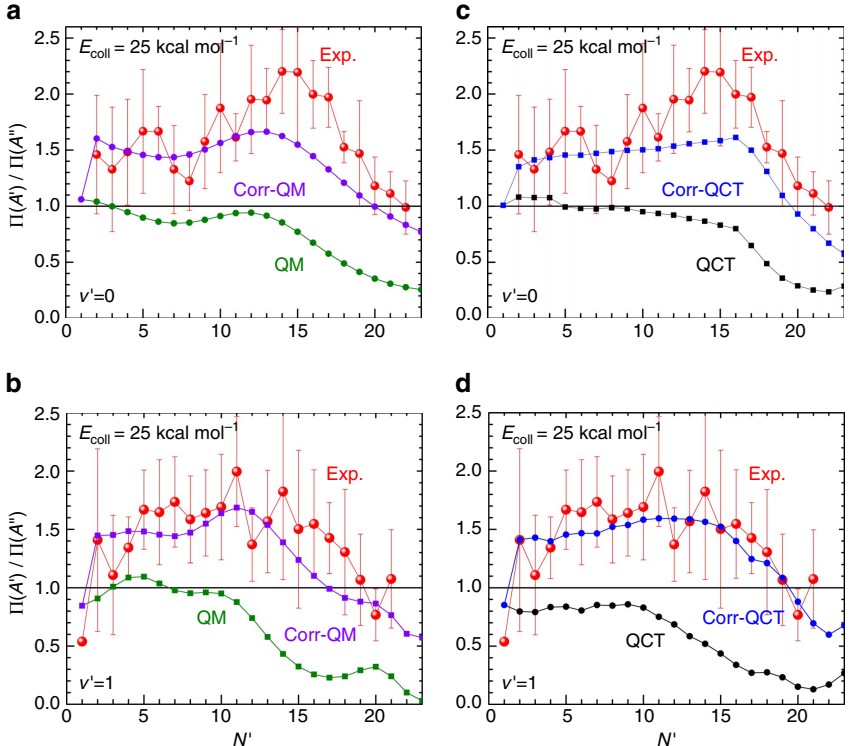

**Figure 3 | $\Lambda$-doublet population ratios for the $O(^3P) + D_2$ reaction at $E_{coll} = 25\,kcal\,mol^{-1}$.** (**a**) QM results for $v' = 0$; (**b**) QM results for $v' = 1$; (**c**) QCT results for $v' = 0$; (**d**) QCT results for $v' = 1$. 'QM' and 'QCT' represent the ratio of the ICSs calculated on the $A'$ and $A''$ PES, whereas 'corr-QM' and 'corr-QCT' are the $\Pi_{A'}/\Pi_{A''}$ ratios after making use of the respective weight factors. The experimental results from ref. 5 are denoted as 'Exp'. The experimental error bars represent the $+1\sigma$ statistical uncertainties based on variations in repeated measurements[5].

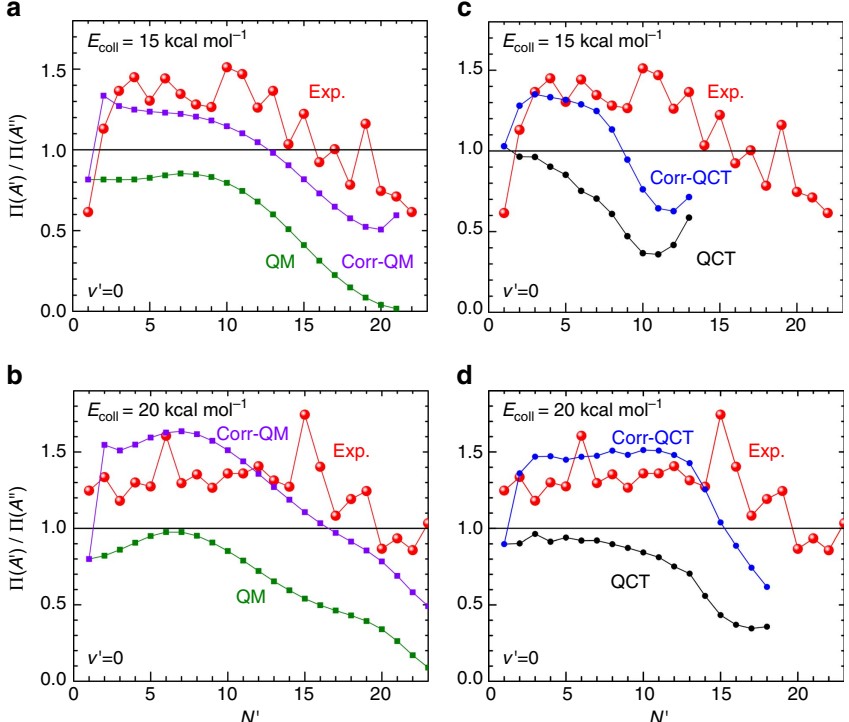

**Figure 4 | Λ-doublet population ratios for the O($^3P$) + D$_2$ reaction at $E_{coll}$ = 15 and 20 kcal mol$^{-1}$.** (a) QM results at $E_{coll}$ = 15 kcal mol$^{-1}$; (b) QM results at $E_{coll}$ = 20 kcal mol$^{-1}$; (c) QCT results at $E_{coll}$ = 15 kcal mol$^{-1}$; (d) QCT results at $E_{coll}$ = 20 kcal mol$^{-1}$; 'QM' and 'QCT' represent the ratio of the ICSs calculated on the A' and A'' PES, whereas 'corr-QM' and 'corr-QCT' are the $\Pi_{A'}/\Pi_{A''}$ ratios after making use of the respective weight factors. The experimental results from ref. 6 are denoted as 'Exp'. No error bars were reported in ref. 6.

experimental Λ-doublet ratio regardless of the collision energy and vibrational manifold studied, uncorrected results fail to account qualitatively for the experimental measurements, predicting a preference towards the $\Pi(A'')$ states.

**Characterization of the mechanisms.** As already discussed, the way in which cross-sections on the A' and A'' PESs are combined to obtain the $\Pi(A')$ and $\Pi(A'')$ populations is strictly related to the alignment of the product rotational angular moment with respect to the recoil direction. To show this effect more clearly, the values of $\sigma(v' = 0, j', \Omega')$ as a function of $\Omega'$ and $j'$ are depicted as gradiational contour maps in Fig. 5 for the A' and A'' PESs. The differences between the respective contour maps are clear to see. The ICS for a given $j'$ state includes the contribution from many $\Omega'$ values on the A'' PES, whereas on the A' the contribution is restricted to relatively few, low $\Omega'$ values. Hence, this picture complements Fig. 2. Negative values of $a_0^{(2)}$ close to the limit imply that $\langle|\Omega'|\rangle$ is very small, nearly 0. If the contributions of higher $\Omega'$ values becomes more significant, the alignment moment tends to be 0.

A more quantitative analysis can be carried out by relating the $\Omega'$ contributions to the weight factors that have been used to extract the Λ-doublet populations. To this end, we have used iso-contour lines for the different values of the weight factors. If in equation (8), $a_0^{(2)}$ is replaced with the $\langle j'\Omega', 20|j'\Omega'\rangle$, which is nothing but the $a_0^{(2)}$ for a pure $(j', \Omega')$ state, we can assign a single weight factor to every point on the $j' - \Omega'$ surface. On the A' PES, most of the reactivity comes from low $\Omega'$ values ($\mathbf{j'} \perp \mathbf{k'}$), falling within the $W_{A'} > 0.75$ limits shown by the central dashed lines in Fig. 5. In contrast, on the A'' PES we find two different trends. For the highest $j'$ values ($j' > 15$) most of the reactivity corresponds to low $\Omega'$, as for the A' PES, although some contributions from higher $\Omega'$ values can also be seen. However,

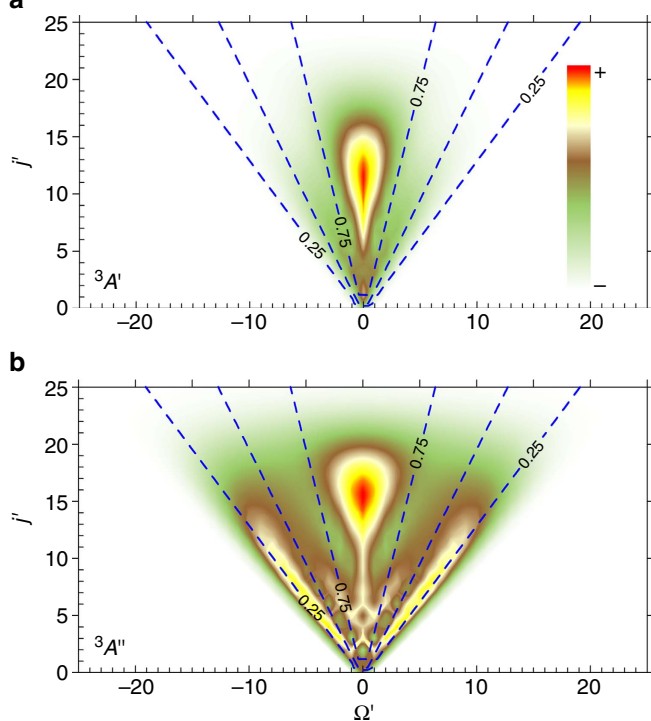

**Figure 5 | Contour plots of the $\Omega'$-resolved QM cross-sections.** (a) $\sigma(v' = 0, j', \Omega')$ calculated on the A' PES as a function of both $\Omega'$ and $j'$. (b) $\sigma(v' = 0, j', \Omega')$ calculated on the A'' PES. The contour lines indicate the values of $W_{A'}$ and $W_{A''}$ for a given combination of $j'$ and $\Omega'$. The data were obtained from the QM calculations at $E_{coll}$ = 25 kcal mol$^{-1}$.

with decreasing $j'$, the low $\Omega'$ peak coexists with additional peaks corresponding to $\Omega' \approx j'$ values ($\mathbf{j'}\|\mathbf{k'}$), which appear along the $W_{A''} \approx 0.25$ dashed lines. The averaging over these two contributions leads to a nearly isotropic alignment ($a_0^{(2)} \sim 0$). These contributions represent two distinct mechanisms: one coplanar that gives rise to low $\Omega'$ and another one that takes place only on the $A''$ PES, which correlates with high $\Omega'$ states and for which the three-atom and OD rotational plane tend to be orthogonal. In the Supplementary Movies 1 and 2, examples of both mechanisms are illustrated with animated trajectories. The main features of these trajectories are discussed in the Supplementary Note 2.

## Discussion

In spite of the tiny energy difference between the $\Lambda$-doublet pair of states, a clear preference towards a particular $\Lambda$-doublet state is observed for many chemical reactions and photodissociation processes. This intriguing fact, whose significance is known to be connected to the reactivity on concurrent PESs and the evolution of the electronic density in chemical processes, has long puzzled researchers and has been the subject of an ongoing discussion for more than 30 years.

In this study, we have presented a method that allows one to extract the $\Lambda$-doublet populations from the cross-sections on the $A'$ and $A''$ PESs. It is shown that the transformation between the reactivities on the $A'$ and $A''$ PESs and the $\Lambda$-doublet populations only requires knowledge about the stereodynamics of the reaction, in particular of the alignment of the product rotational angular momentum, $\mathbf{j'}$, along the recoil direction, $\mathbf{k'}$.

This method has been applied to the $O(^3P) + D_2$ reaction, for which we have carried out QCT and QM adiabatic calculations. Although couplings between the concurrent PES have not been included, our method accounts quantitatively for the experimental $\Lambda$-doublet populations obtained by Minton and colleagues[5,6] that have thus far remained unexplained. The analysis of the results has shown that the preference for the $\Pi(A')$ $\Lambda$-doublet state is due to the existence of an additional mechanism on the $A''$ PES for which the OD rotational plane tends to be orthogonal to the three-atom plane. This mechanism has been traced back to the comparative topographies of the $A'$ and $A''$ PESs, the latter characterized by a broader cone of acceptance.

The implications of the results and methodology presented in this work go beyond the reproduction of experimental measurements, as they reveal the underlying connection between the reaction mechanism and the population of the $\Lambda$-doublet states. The strategy developed to quantify such a connection is, in principle, general and can be used in combination with scattering data obtained using both QCT and QM adiabatic and non-adiabatic approaches. The methodology might also be extended to more complex reactions generating open shell radicals, such as $H + CO_2$ (ref. 28), $H + N_2O$ (ref. 29) and $O(^3P) + CH_4$ (ref. 30), whose lambda-doublet propensities have been measured. In addition, the scheme presented here could be inverted so as to assess the role of the different PESs that participate in a process from the experimentally measured relative populations of the $\Lambda$-doublet states and the polarization parameters.

## Methods

**Ab initio calculations.** The PESs of the lowest $1^3A'$ and $1^3A''$ states were determined using 3,500 ab initio points for each PES that were calculated using the MOLPRO suite of programmes[31,32]. For both oxygen and hydrogen atoms, an aug-cc-pV5Z basis set including spdfg basis functions was used. To obtain an accurate and homogeneous description of the PESs, the state-average complete active space self-consistent field method[33] was employed. The active space considered consisted of eight electrons distributed in six orbitals ($2–6a'$ and $1a''$), to include all valence orbitals of oxygen and the $1s$ orbitals from both hydrogen atoms. The state-average orbitals and multireference configurations obtained were then used to calculate both the lowest $1^3A'$ and the lowest $1^3A''$ state energies with

the internally contracted multireference configuration interaction method, including single and double excitations[34] and the Davidson correction[35].

The ab initio internally contracted multireference configuration interaction + Q energies for the $1^3A'$ and $1^3A''$ electronic states were fitted separately using the GFIT3C procedure introduced in refs 36–38, in which the global PES is represented by a many-body expansion:

$$V_{ABC} = \sum_A V_A^{(1)} + \sum_A V_{AB}^{(2)}(r_{AB}) + V_{ABC}^{(3)}(r_{AB}, r_{AC}, r_{BC}), \qquad (9)$$

where $V_A^{(1)}$ represents the energy of the atoms (A = O,H,H) in the ground electronic state, $V_{AB}^{(2)}$ the diatomic terms (AB = OH,OH,HH) and $V_{ABC}^{(3)}$ the three-body term (ABC = OHH). This fitting procedure allows one to obtain a smooth analytical form of the potential and its gradient. The overall root-mean-square error of the two analytical potentials calculated over the 3,500 geometries was found to be 0.61 and 0.44 kcal mol$^{-1}$ for the $1^3A'$ and $1^3A''$ states, respectively.

In what follows, we will compare the PESs devised in this work with the benchmark PESs obtained by Rogers et al.[24] (hereinafter RWKW PESs). Both sets of PESs are qualitatively similar and are characterized by a collinear barrier of height $\approx 0.6$ eV, a larger 'cone of acceptance' on the $^3A''$ PES and lack deep potential wells. Our PESs show a T-shape Van der Waals well in the entrance channel for both the $A'$ and $A''$ states. They do not appear on the RWKW PESs, as they fitted their surfaces using a cutoff and a smooth damping in the long-range region.

The geometry and energy of the saddle points for both sets of PESs are presented in Table 1 and compared with the ab initio values calculated using a the aug-cc-pV5Z basis set. The barriers are slightly larger on our PESs ($\sim 20$ meV) and reproduce slightly better the energetic degeneracy of the saddle points. This fact is important, because the breaking of such degeneracy may increase artificially the reactivity on one the PESs and influence the calculated $\Lambda$-doublet propensities.

In Supplementary Fig. 4 we show the QM state-to-state reactive cross-sections calculated on the $A'$ and $A''$ PESs as a function of the products' internal energy at 25 kcal mol$^{-1}$, similar to that published in ref. 39. As can be seen, the RWKW PESs[24] are more reactive due to their slightly smaller barrier. The main differences between the reactive cross-sections are observed on the $A''$ PES in the 0.6–0.8 eV internal energy range, where a bump is observed for the RWKW PESs. However, and in spite of the aforementioned differences, the $\Lambda$-doublet ratios predicted using both set of PESs are almost identical as it is shown in Supplementary Fig. 5, owing to the fact that the relatively small differences in rotationally resolved cross-sections are compensated when the ratios are calculated. These results lend additional credence to the methodology devised in this study, showing that the preference observed towards the $\Pi(A')$ state is not caused by small details of the PES but is determined by the overall reaction mechanism.

**Dynamical calculations.** Based on the PESs obtained, QCT and time-independent QM calculations were carried out at the three collision energies: 15, 20 and 25 kcal mol$^{-1}$. QCT calculations consisted of batches of $5 \times 10^6$ trajectories following the methodology described in refs 22,23. The trajectories were started and finished at a atom–diatom distance of 20 a.u. ($\sim 10$ Å) and the integration step size was chosen to be 0.05 fs, which guarantees a energy conservation better than a 0.01%. The rovibrational energy of the reactant $D_2$ molecule was calculated by semiclassical quantization of the action using the potential given by the asymptotic reactant valley of the PES. The assignment of the product quantum numbers was carried out by equating the square of the classical $D_2$ molecule rotational angular momentum to $j'(j'+1)\hbar^2$. The vibrational quantum number $v'$ was found by equating the internal energy of the products to a rovibrational Dunham expansion. The 'quantum numbers' so obtained were rounded to the nearest integer.

To extract the contributions of each trajectory to the $\Pi(A')$ or $\Pi(A'')$ $\Lambda$-doublet states, it is sufficient to determine the classical product polarization parameters $a_0^{(2)}$ with respect to the recoil direction on each PES. This polarization parameter is given by $\langle P_2(cos\theta_{\mathbf{j'k'}})\rangle$, where the brackets indicate the averaging over the set of reactive trajectories leading to a given final state.

Time-independent QM calculations were carried out using the ABC[21] code. The basis set for the calculations included all the diatomic energy levels up to 63.4 kcal mol$^{-1}$. The propagation was carried out in 150 log-derivative sectors up to a distance of 20 a.u. For $J>0$, the value of $\Omega_{max}$, the maximum value of the projection of $J$ and the rotational angular momentum onto the body fix axis was

---

**Table 1 | Equilibrium structures and energies at the saddle point for the current PESs and the benchmark RWKW PESs obtained by Rogers et al.[24]**

|  | Current PES | | RWKW PES | | aug-cc-pV5Z |
|---|---|---|---|---|---|
|  | $^3A''$ | $^3A'$ | $^3A''$ | $^3A'$ | $^3\Pi$ |
| $r_{OH}$ (a.u.) | 2.295 | 2.281 | 2.300 | 2.309 | 2.301 |
| $r_{HH}$ (a.u.) | 1.680 | 1.709 | 1.705 | 1.705 | 1.681 |
| $E$(eV)* | 0.588 | 0.589 | 0.565 | 0.573 | 0.591 |

*The energies are relative to the reactants asymptote.

always chosen to be larger than the maximum value of $j'$ energetically accessible ($\Omega_{max} = 30$).

**Classical and semiclassical deduction of equation (5).** The $A'$ or $A''$ electronic symmetry of the PES is defined with respect to the rotating body-fixed DOD plane, defined by $\mathbf{r}'$ and $\mathbf{R}'$ (OD internuclear vector and the atom-diatom D–OD vector, respectively). In turn, the symmetry of the $\Lambda$-doublet states is defined with respect to the reflection in the OD rotation plane that contains $\mathbf{r}'$ and is perpendicular to $\mathbf{j}'$, the nuclear rotational angular momentum.

Therefore, the relevant angle is $\theta_{j'u}$, that is, the angle between $\mathbf{j}'$ and $\mathbf{u}$, a vector in the direction of $\mathbf{r}' \times \mathbf{R}'$. The vector $\mathbf{R}'$ is asymptotically parallel to the product recoil vector $\mathbf{k}'$ and hereinafter we will use the latter as reference. In fact, $\theta_{j'u}$ represents the dihedral angle between the molecular plane, and the three-atom plane. As pointed out in refs 15,19, $\cos^2\theta_{j'u}$ can be used to relate the symmetry of the $\Lambda$-doublet levels to the symmetry of the PES.

The use of $\cos^2\theta_{j'u}$ stems from the fact that it represents the probability that for each PES, the OD molecule will be produced in a given $\Lambda$-doublet state. Classically, the use of the square of the cosine of $\theta_{j'u}$ can be justified as we are interested in the mutual alignment of the planes depicted in the Supplementary Fig. 1.

Without any loss of generality, we can select a space-fixed (scattering) frame of coordinates in which both the product rotational angular momentum, $\mathbf{j}'$, and the recoil direction, $\mathbf{k}'$, are fixed. Let us also assume that $\mathbf{j}'$ lies along the $z$ axis (see Supplementary Fig. 1) and $\mathbf{k}'$ is contained in the $xz$ plane. With this choice, the OD internuclear axis, $\mathbf{r}'$, will lie in the $xy$ plane. Then, $\theta_{k'j'}$, the angle between $\mathbf{j}'$ and $\mathbf{k}'$, is given by:

$$\cos\theta_{k'j'} = \frac{k'_z}{|\mathbf{k}'|}, \tag{10}$$

where $k'_z$ is the $z$ component of $\mathbf{k}'$. Since $\mathbf{u} = \mathbf{r}' \times \mathbf{k}'$, $|\mathbf{u}| = |\mathbf{r}'||\mathbf{k}'|\sin\theta_{k'r'}$, where $\theta_{k'r'}$ is the angle between $\mathbf{k}'$ and $\mathbf{r}'$. Hence, the cosine of the angle between $\mathbf{j}'$ and $\mathbf{u}'$ is given by:

$$\cos\theta_{j'u} = \frac{u_z}{|\mathbf{u}|} = -\frac{1}{|\mathbf{u}|}r'_y k'_x = -\frac{|\mathbf{r}'||\mathbf{k}'|}{|\mathbf{u}|}\sin\phi_{r'}\sin\theta_{k'j'}$$
$$= -\frac{\sin\theta_{k'j'}\sin\varphi_{r'}}{\sin\theta_{k'r'}},$$

where $\phi_{r'}$ is the azimuthal angle of $\mathbf{r}'$.

Using the law of cosines, it can be shown that:

$$\cos\theta_{k'r'} = \cos\theta_{k'j'}\cos\theta_{j'r'} + \sin\theta_{k'j'}\sin\theta_{j'r'}\cos\phi_{r'} = \sin\theta_{k'j'}\cos\phi_{r'}, \tag{12}$$

where we have used the fact that $\mathbf{r}'$ is perpendicular to $\mathbf{j}'$. Therefore, combining equations (11) and (12) we obtain the following expression for $\cos^2\theta_{j'u}$:

$$\cos^2\theta_{j'u} = \frac{\sin^2\theta_{k'j'}(1 - \cos^2\phi_{r'})}{1 - \sin^2\theta_{k'j'}\cos^2\phi_{r'}}. \tag{13}$$

In the chosen space-fixed reference frame, $\mathbf{j}'$ and $\mathbf{k}'$ do not change with rotation in the product asymptote for a given trajectory. Hence, the only variable that changes with rotation in equation (13) is $\phi_{r'}$. Averaging over a rotational period, that is, integrating equation (13) over $\phi_{r'}$ and dividing by $2\pi$, the average value of $\cos^2\theta_{j'u}$ over a rotational period is given by:

$$\langle\cos^2\theta_{j'u}\rangle_{rot} = 1 - \left|\cos^2\theta_{k'j'}\right|^{1/2}. \tag{14}$$

Equation (14) is particularly relevant, as it relates $\theta_{j'u}$, the angle between the normal vectors to the OD rotation plane and to the three-atom plane, with $\theta_{k'j'}$. Semiclassically, $|\mathbf{j}'|\cos\theta_{k'j'}$ is the projection of $\mathbf{j}'$ onto $\mathbf{k}'$, that is, the product helicity $\Omega'$, and the right-hand side of equation (5) is recovered.

**QM deduction of equation (5).** Following refs 10,13,40, the OH molecular rotational wave function can be written as

$$|j'\Omega'\Lambda'\rangle = \left(\frac{2j'+1}{4\pi}\right)^{1/2} D^{j'*}_{\Omega'\Lambda'}(\alpha, \beta, \gamma = 0), \tag{15}$$

where $D^{j'*}_{\Omega'\Lambda'}$ is the rotation matrix element and $(\alpha, \beta, \gamma)$ are the Euler angles, which specify the orientation of the body-fixed frame (BF), $xyz$, with respect to the space-fixed (SF) frame, $XYZ$. The BF frame is chosen with $z$ along the OH internuclear axis, $\mathbf{r}'$, whereas the $Z$ axis in the SF frame is chosen along the recoil velocity vector, $\mathbf{k}'$. With this choice, $\beta = \theta_{k'r'}$ and $\alpha = \varphi_{r'}$, which define the direction of $\mathbf{r}'$ in the SF frame. The angle $\gamma$ is chosen to be 0, such that the line of nodes (the intersection of the $XY$ and $xy$ planes) is $y \equiv \mathbf{u}$, which is perpendicular to both $z = \mathbf{r}'$ and $Z = \mathbf{k}'$ and, as discussed in the previous subsection, defines the normal vector to the three-body plane. The projection of the rotational angular momentum, $\mathbf{j}'$, along the BF $z$ axis is $\Lambda'$. If the open-shell character of the molecule is neglected, $\Lambda' = 0$. In turn, $\Omega'$ is the projection of $\mathbf{j}'$ along the SF $Z$ axis (usually $m'$ is used to designate the projection of $\mathbf{j}'$ onto the SF axis, but in the present case, as $\mathbf{k}'$ is taken as $Z$, it corresponds to the helicity, which is commonly designated by $\Omega'$). As discussed in refs 10,13,14, with this choice of frames, for $\Omega' = 0$ and $j' \gg 1$, $\mathbf{j}'$ lies along $\mathbf{u}$ (the $y$ axis) and $xz$ is the rotation plane. For $\Omega' = j'$ and $j' \gg 1$, $\mathbf{j}'$ is along the $x$ axis, which in this case is along $-Z$ and the rotation plane is $yz$.

Classically, $|\mathbf{j}'|^2\cos^2\theta_{j'\mathbf{u}}$ represents the square of the projection of $\mathbf{j}'$ onto the $\mathbf{u}$ vector. In QM, the equivalent magnitude would be $\langle\hat{j}_u^2\rangle$, the expectation value of the square of the operator that represents the projection along $\mathbf{u}$. It can be shown that the expression of $\hat{j}_u$ is simply $-i\partial/\partial\theta_{k'r'}$ (ref. 41). Therefore,

$$\langle\hat{j}_u^2\rangle = \langle j'\Omega'\Lambda'|\hat{j}_u^2|j'\Omega'\Lambda'\rangle$$
$$= \frac{2j'+1}{4\pi}\int_0^{2\pi}\int_{-1}^{1}D^{j'}_{\Omega'\Lambda'}(\varphi_{r'}, \theta_{k'r'}, 0) \tag{16}$$
$$\cdot\left(-\partial^2/\partial\theta^2_{k'r'}\right)D^{j'*}_{\Omega'\Lambda'}(\varphi_{r'}, \theta_{k'r'}, 0)\mathrm{d}\varphi_{r'}\,\mathrm{d}\cos\theta_{k'r'}$$

It can be shown that the result of this integral is almost exactly

$$\langle\hat{j}_u^2\rangle = j'\left(j'+1 - \delta_{\Omega',0}\right)\left(1 - \left|\frac{\Omega'^2}{j'(j'+1)}\right|^{1/2}\right) \tag{17}$$

where $\delta_{\Omega',0}$ stems from the fact that for $\Omega' = 0$ the maximum value of the projection is $j'$. Apart from this correction, this equation is the semiclassical expression equation (5) of the main text.

**Data availability.** The authors declare that all data supporting the findings of this study are available from the corresponding author upon request.

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

## Acknowledgements

We acknowledge funding by the Spanish Ministry of Economy and Competitiveness (grants CTQ2012-37404-C02, CTQ2015-65033-P and Consolider Ingenio 2010 CSD2009-00038). M.B. gratefully acknowledges the support of the U.K. EPSRC (via Programme Grant EP/L005913/1). P.G.J. acknowledges the Spanish Ministry of Economy and Competitiveness for the Juan de la Cierva fellowship (IJCI-2014-20615). A.Z. acknowledges the European Research Council under the European Union's Seventh Framework Programme (FP/2007-2013)/ERC Grant Agreement number 610256 (NANOCOSMOS).

## Author contributions

P.G.J., F.J.A. and M.B. devised the method. P.G.J., A.Z. and J.A. carried out the calculations. P.G.J. and F.J.A. wrote the paper with contributions from all the authors.

## Additional information

**Competing financial interests:** The authors declare no competing financial interests.

