## [Peer Review File · Nature Communications]

Reviewer #1 (Remarks to the Author):

This paper presents a theoretical analysis and calculations aimed at understanding the lambda-doublet populations produced in the reaction of $O(^3P)$ with D_2 . The paper is very well motivated and clearly written, and the new theoretical results do reproduce experimental lambda-doublet populations quite well. The authors present both qualitative and quantitative results, which provides the reader with a good picture of how the results come to be. The graphics add to the story in a helpful manner. Overall, this is a very lucid account that presents new high-quality results that will be interesting to many readers. I believe it should be accepted for Nature Communications. If this were a specialty journal, my comments would end there. However, for a more general publication like Nature Communications, I would appreciate a statement in the conclusions of the more general implications of the results: okay, you have explained these results in the title system, but how can this understanding be used going forward, how are the ideas manifest in more complex systems, or what more general insight is now revealed? As it is, the conclusions end a little abruptly.

Reviewer #3 (Remarks to the Author):

This paper describes a new method to predict the lambda-doublet ratios in the products of 3-atom exchange reactions. The paper is very clearly written, with carefully drafted, illustrative figures.

As the introduction correctly states, the classical nuclear dynamics of such apparently simple systems are largely well-understood, but there remain fundamental challenges to explaining the detailed branching between near-degenerate electronic states of different symmetry. These so-called lambda states can be produced in very uneven ratios. It has been known for some time that these are connected to, and provide a signature of, the potential energy surfaces of different electronic symmetry over which the reaction can proceed. The important general advance presented here is to derive a new but straightforward approach (encapsulated in eq(5)) which describes how well the symmetry of an intermediate surface is retained in the preferred lambda-doublet of the final diatomic product in terms of the correlation between the final translational recoil direction (k') and the product rotational angular momentum (j').

This approach is completely general and can be applied to any 3-atom reactive system. In this respect the paper is already of considerable interest to the field of reaction dynamics. The impact of the paper is enhanced even further, however, by its application to the prototypical first-row 3-atom system, $O(^3P) + D_2 \rightarrow OD + D$. Experimental measurements of the OD lambda-doublet ratios in this system were published relatively recently by Minton, McKendrick and co-workers (originally in Nature Chemistry, then a fuller account in JACS). The very surprising result was reported that the A' lambda-doublet was the dominant product, despite the prior theoretical prediction that the integral cross section was higher on the $3A''$ surface. A commentary by Alexander in a co-published Nature Chemistry News and Views article emphasized how interesting and currently unexplained this observation was, setting a challenge to theory. That challenge has been taken up and met fully here. Jambolina et al. show, in essence, that the different stereodynamics on the $3A'$ and $3A''$ surfaces lead to a higher degree of preservation of the intermediate symmetry for the $3A'$ than the $3A''$ surface. This outweighs the higher reactivity on the $3A''$ surface, rationalising the previously inexplicable result. The experimental j' -dependent lambda-doublet ratios at different collision energies are reproduced near-quantitatively. This is a truly beautiful result which is certain to attract considerable interest.

I therefore recommend strongly that the paper is suitable for publication in Nature Communications. I have only two suggestions that the authors might wish to address prior to publication:

(1) New ab initio $3A'$ and $3A''$ potential-energy surfaces were calculated as part of this work. It is

only implicit in the text, when discussing the known difference in these PESs with bending angle, that the new surfaces are relatively similar to the previous benchmark calculations in ref [23]. There is also no comparison with previous PESs in the Computational Methods section. A brief commentary describing the extent of quantitative similarity would be helpful.

(2) Similarly, there have been previous calculations of the reaction cross sections on the previous 3A' and 3A" surfaces (Garton et al. *J. Phys. Chem. A* 2006, 110, 1327) by both quantum and QCT methods. It would be helpful to state briefly the level of agreement with the new results here (i.e. with the top left panel of Figure 1).

Response to the Reviewers

Reviewer # 1

This paper presents a theoretical analysis and calculations aimed at understanding the lambda-doublet populations produced in the reaction of O ³P with D₂. The paper is very well motivated and clearly written, and the new theoretical results do reproduce experimental lambda-doublet populations quite well. The authors present both qualitative and quantitative results, which provides the reader with a good picture of how the results come to be. The graphics add to the story in a helpful manner. Overall, this is a very lucid account that presents new high-quality results that will be interesting to many readers. I believe it should be accepted for Nature Communications. If this were a specialty journal, my comments would end there.

We are grateful to reviewer #1 for their positive appraisal.

However, for a more general publication like Nature Communications, I would appreciate a statement in the conclusions of the more general implications of the results: okay, you have explained these results in the title system, but how can this understanding be used going forward, how are the ideas manifest in more complex systems, or what more general insight is now revealed? As it is, the conclusions end a little abruptly.

We have added the following paragraph at the end of the conclusions where we highlight the general insight that our methodology provide, discuss its application for the study of polyatomic reactions and how to obtain the reactivity on concurrent PESs based on experimental measurements of the cross sections and polarization parameters.

The implications of the results and methodology presented in this work go beyond the reproduction of experimental measurements since they reveal the underlying connection between the reaction mechanism and the population of

the Λ -doublet states. The strategy developed to quantify such connection is, in principle, general and can be used in combination with scattering data obtained using both QCT and QM adiabatic and non-adiabatic approaches. The methodology might also be extended to more complex reactions generating open shell radicals, such as $\text{H} + \text{CO}_2$,²⁹ $\text{H} + \text{N}_2\text{O}$,³⁰ and $\text{O}(^3\text{P}) + \text{CH}_4$ ³¹ whose lambda-doublet propensities have been measured. Additionally, the scheme here presented could be inverted so as to assess the role of the different PESs that participate in a process from the experimentally measured relative populations of the Λ -doublet states and the polarization parameters.

On top of this, we have added one sentence in the introduction to stress that Λ -doublet propensity is common not only in chemical reactions but also in other important molecular processes.:

“In spite of the tiny energy difference between the Λ -doublet pair of states, a clear preference towards one of them is observed in many chemical reactions, inelastic collisions and photodissociation processes^{1–6} and even as the origin of OH astronomical masers.^{7,8}”

Reviewer # 3

This paper describes a new method to predict the lambda-doublet ratios in the products of 3-atom exchange reactions. The paper is very clearly written, with carefully drafted, illustrative figures.

As the introduction correctly states, the classical nuclear dynamics of such apparently simple systems are largely well-understood, but there remain fundamental challenges to explaining the detailed branching between near-degenerate electronic states of different symmetry. These so-called lambda states can be produced in very uneven ratios. It has been known for some time that these are connected to, and provide a signature of, the potential energy surfaces of different electronic symmetry over which the reaction can proceed. The important general advance presented here is to derive a new but straightforward approach (encapsulated in eq(5)) which describes how well the symmetry of an intermediate surface is retained in the preferred lambda-doublet of the final diatomic product in terms of the correlation between the final translational recoil direction (k') and the product rotational angular momentum (j').

This approach is completely general and can be applied to any 3-atom reactive system. In this respect the paper is already of considerable interest to the field of reaction dynamics. The impact of the paper is enhanced even further, however, by its application to the prototypical first-row 3-atom system, $\text{O}(^3\text{P}) + \text{D}_2 \rightarrow \text{OD} + \text{D}$. Experimental measurements of the OD lambda-doublet ratios in this system were published relatively recently by Minton, McKendrick and co-workers (originally in Nature Chemistry, then a fuller account in JACS). The very surprising result was reported that the A' lambda-doublet was the dominant product, despite the prior theoretical prediction that the integral cross section was higher on the $3A''$ surface. A commentary by Alexander in a co-published Nature Chemistry News and Views article emphasized how interesting and currently unexplained this observation was, setting a challenge to theory. That challenge has been taken up and met fully here. Jambolina

et al. show, in essence, that the different stereodynamics on the 3A' and 3A'' surfaces lead to a higher degree of preservation of the intermediate symmetry for the 3A' than the 3A'' surface. This outweighs the higher reactivity on the 3A'' surface, rationalising the previously inexplicable result. The experimental j' -dependent lambda-doublet ratios at different collision energies are reproduced near-quantitatively. This is a truly beautiful result which is certain to attract considerable interest.

I therefore recommend strongly that the paper is suitable for publication in Nature Communications.

We thank the reviewer for their praises and positive review.

I have only two suggestions that the authors might wish to address prior to publication:

(1) New ab initio 3A' and 3A'' potential-energy surfaces were calculated as part of this work. It is only implicit in the text, when discussing the known difference in these PESs with bending angle, that the new surfaces are relatively similar to the previous benchmark calculations in ref [23]. There is also no comparison with previous PESs in the Computational Methods section. A brief commentary describing the extent of quantitative similarity would be helpful.

Following the reviewer's comments we have added the following paragraph to the method section where we compare the similarity between our set of PESs and those calculated by Roger's et al. (RWKW PES).

“In what follows we will compare the PESs devised in this work with the benchmark RWKW PESs, obtained by Rogers et al.²⁵ Both sets of PESs are qualitatively similar and are characterized by a ~ 0.6 eV collinear barrier, a larger “cone of acceptance” on the A'' PES and lack deep potential wells. Our PESs show a T-shape Van der Waals well in the entrance channel for both the A' and A'' states. They do not appear on the RWKW PESs since they fitted their surfaces using a cut-off and a smooth damping in the long range region.

The geometry and energy of the saddle points for both sets of PESs are presented in Table 1 and compared with the ab-initio values calculated using a the aug-cc-pV5Z basis set. The barriers are slightly larger on our PESs (20 meV) and reproduce slightly better the energetic degeneracy of the saddle-points. This fact is important because the breaking of such degeneracy may increase artificially the reactivity on one the PESs and influence the calculated Λ -doublet propensities.”

We have also included a new table (Table I) where the energy and geometry of the saddle points for both set of PESs are displayed.

(2) Similarly, there have been previous calculations of the reaction cross sections on the previous 3A' and 3A'' surfaces (Garton et al. J. Phys. Chem. A 2006, 110, 1327) by both quantum and QCT methods. It would be helpful to state briefly the level of agreement with the new results here (i.e. with the top left panel of Figure 1).

Following the reviewer's suggestions we have included one additional section in the Supplementary Information, which is referred in the main text, where we compare our results using the new PES with the previous results calculated on the benchmark RWKW PES by Rogers et al. This comparison is illustrated through Fig. S6.

Fig. S6 shows the QM state-to-state reactive cross sections calculated on the A' and A'' PESs as a function of the products internal energy at 25 kcal/mol, similarly to that published in Ref. 11. As can be seen, RWKW PESs are more reactive due to their slightly smaller barrier (see Methods section in the main text for a brief discussion).

The main differences between the reactive cross sections are observed on the A'' PES in the 0.6-0.8 eV internal energy range, where a bump is observed for Rogers et al PES.

Moreover, we have added one additional figure (see below) where we show that the Λ -doublet ratios calculated using both sets of PESs are almost identical, which lends additional credence to our results.

This figure is commented in the Supplementary Information as follows:

“...the Λ -doublet ratios predicted using both sets of PESs are almost identical as shown in Fig. S7 due to the fact that the relatively small differences in rotationally resolved cross sections are compensated when the ratios are calculated. These results lend additional credence to the methodology devised in this article showing that the preference observed towards the $\Pi(A')$ state is not caused by small details of the PES but is determined by the overall reaction mechanism.”

REVIEWERS' COMMENTS:

Reviewer #1 (Remarks to the Author):

It appears the authors have addressed both my comments and those of the other referee. I believe this is a very nice contribution and that it should be accepted for publication.

Reviewer #3 (Remarks to the Author):

I was Reviewer #3 in the original submission.

The authors have responded very thoroughly and appropriately to my requests to:

1) provide more details of the comparison between their new PESs and the existing version by Rogers et al.

and

2) compare the results of the scattering calculations done on their new surfaces with those by Schatz and co-workers on the Rogers et al. surface.

The degree of similarity in both cases reinforces the conclusion that the phenomena they describe are not specific to details of any particular form of the potential.

I believe the remarks they have added in response to Reviewer #1 will also improve the accessibility to a more general audience.

I am very happy to recommend publication of the revised version.